# Acquisition of Humoral Immune Responses in Convalescent Japanese People with SARS-CoV-2 (COVID-19) Infection in 2021

**DOI:** 10.3390/v15091842

**Published:** 2023-08-30

**Authors:** Koshiro Monzen, Takanori Watanabe, Toshihiro Okabe, Hisakuni Sekino, Hironori Nakagami, Ryuichi Morishita

**Affiliations:** 1Shinjuku Tsurukame Clinic, 2-11-15 Yoyogi, Shibuya-ku, Tokyo 151-0053, Japan; 2Uehonmachi Watanabe Clinic, 1-15 Uenomiya-cho, Tennoji-ku, Osaka 543-0037, Osaka, Japan; 3Ueda Hospital, 1-7-1 Inazu-cho, Toyonaka 561-0854, Osaka, Japan; 4Sekino Hospital, 3-28-3 Ikebukuro, Toshima-ku, Tokyo 171-0014, Japan; 5Department of Health Development and Medicine, Graduate School of Medicine, Osaka University, 2-2 Yamada-oka, Suita 565-0871, Osaka, Japan; 6Department of Clinical Gene Therapy, Graduate School of Medicine, Osaka University, 2-2 Yamada-oka, Suita 565-0871, Osaka, Japan

**Keywords:** COVID-19, SARS-CoV-2, convalescent serum, neutralizing antibody

## Abstract

We investigated humoral immune responses in 222 unvaccinated Japanese people after recovery from severe acute respiratory syndrome coronavirus 2 (SARS-CoV-2) infection in 2021. Anti-spike-protein IgG antibody levels and neutralizing antibody titers were measured in serum samples obtained within 20–180 days after diagnosis. The geometric mean of antibody titers was 1555 ELU/mL (95% confidence interval (CI) = 1257–1923), and the neutralizing activity (50% inhibitory dilution) was 253 (95% CI = 204–313). The antibody titer and neutralizing activity both increased with increasing disease severity, and both values were approximately fourfold higher for hospitalized patients than for non-hospitalized patients. However, these differences were smaller in older patients. The humoral immune response, which increased with increasing disease severity, gradually decreased over time after SARS-CoV-2 infection. Most patients with mild or moderate symptoms sustained neutralizing activity for up to 180 days after the infection; the decay of the neutralizing activity in the asymptomatic patients was rather faster than in the other groups. Around 11.7% (26/222) of patients had very low neutralizing activity, and half of these were aged in their 20s. Our study’s results show the importance of measuring the neutralizing activity to confirm the immune status and to estimate the timing of vaccines.

## 1. Introduction

Coronavirus disease 2019 (COVID-19) is an infectious disease caused by severe acute respiratory syndrome coronavirus 2 (SARS-CoV-2), which was first identified in 2019 in Wuhan, China [1]. The outbreak of COVID-19 began with “pneumonia of unknown cause” in Wuhan, Hubei Province, in December 2019, followed by the international spread of the disease, mainly in China. The World Health Organization (WHO) declared a Public Health Emergency of International Concern (PHEIC) on 30 January 2020, and as of 18 March 2021, the WHO had reported 120,383,919 cases and 2,664,386 deaths worldwide, with 223 countries/regions having confirmed cases. Multiple coronavirus variants (e.g., Alpha, Beta, Gamma, Delta, and Omicron) have been discovered to date, and they have spread globally (WHO).

It remained difficult to control SARS-CoV-2 transmission during pandemic waves because of the large number of people with asymptomatic infection, who had similar viral loads to those of symptomatic patients, as well as viral shedding in symptomatic patients before symptom onset. In Japan, seven SARS-CoV-2 epidemic waves were encountered through the middle of 2022. Each wave consisted of a sharp surge and subsequent decline in new infection cases. Some reports described the acquisition of humoral immune responses after recovery from SARS-CoV-2 infection in Japanese patients [2,3]; D614G-mutation-carrying variants were the primary causes of COVID-19 [4].

Many important reports on the relationship between antibody production and patient background have been accumulated. In general, neutralizing antibody activities in patients with COVID-19 vary widely from patient to patient; peak neutralizing antibody activities, however, are known to increase in proportion to the severity of the disease [5]. However, the well-balanced and coordinated function of antibody-producing B cells and T cells is important for biological defense during acute infection with novel coronaviruses [6], and the importance of the natural immune system has been pointed out, which contributes to an asymptomatic recovery despite the low production of antibodies [7]. Thus, a complex network of collaboration and communication among cells functioning to eliminate the virus should be considered; however, the information regarding antibody levels and activity analyzed by patient background and the decay of neutralizing activity remains insufficient.

To manage the inoculation strategy for the available coronavirus vaccines, it is considered to be important for Japanese medical personnel to know the naturally acquired post-SARS-CoV-2 infection immunity status of patients and compare it with coronavirus-vaccine-derived immunity. In this study, we evaluated the anti-spike and neutralizing antibody levels of patients infected with SARS-CoV-2 between January 2021 and September 2021. During this period, the Alpha, Delta, and D614G-mutation-carrying variants were the primary causes of COVID-19 [4]. The objective of this study was to obtain information for the development of new prophylactic vaccines and new treatments against COVID-19 in Japan.

## 2. Materials and Methods

### 2.1. Study Samples

#### 2.1.1. Inclusion Criteria

Serum samples from non-hospitalized individuals or hospitalized individuals who had recuperated from previous SARS-CoV-2 infection, satisfying all of the following inclusion criteria, were selected in this observational study:Samples from study subjects from whom written voluntary informed consent to participate in this study was obtained.Samples from study subjects who gave consent for the principal investigator, etc., to collect information on the diagnosis, treatment, etc., from the medical institution, health center, etc., where the SARS-CoV-2 infection was diagnosed/treated.Samples from study subjects aged 20 years or older at the time of informed consent.Samples from study subjects who had recuperated from infection after testing positive for SARS-CoV-2 diagnosed through nucleic acid detection or antigen testing, and who tested negative on nucleic acid detection or antigen tests, or archival samples from individuals who had recuperated and were similar to those mentioned above.Samples from 20 to 180 days after testing positive for SARS-CoV-2.

#### 2.1.2. Exclusion Criteria

Samples were excluded from the analysis if met any of the following criteria:Samples from individuals who had not recuperated from SARS-CoV-2 infection.Samples from recipients of prophylactic COVID-19 vaccines (including products in development).Samples for which there was a request to withdraw consent.Samples considered ineligible by the principal investigator or sub-investigator.

### 2.2. Study Procedure

The SARS-CoV-2 infection was confirmed by nucleic acid detection or antigen testing. Samples from study subjects who had recuperated from infection or archival samples from individuals who had recuperated were transported to a testing facility in order to measure neutralizing activity against the pseudovirus of SARS-CoV-2 and SARS-CoV-2 spike (S) glycoprotein-specific antibody titers. For a sample collected from a study subject, a nucleic acid detection test or antigen test was performed again to confirm the SARS-CoV-2 infection status at the time of sample collection. If negative results were obtained, the data were included in the analysis set of this study, and if positive results were obtained, the data were excluded from the analysis set. Even if archival samples were used, data from samples with negative results were included in the analysis set.

The following information was entered into the case report form for each study subject: anonymized study ID, age, sex, onset date, date of definitive diagnosis, definitive diagnosis result (nucleic acid detection or antigen test), non-hospitalization/hospitalization, severity of SARS-CoV-2 infection, result of assessment of recuperation from infection (nucleic acid detection or antigen test), and date of sample collection. The severity was classified into five levels as follows: asymptomatic, mild (no respiratory symptoms or cough only, without dyspnea; SpO_2_ ≥ 96%; no evidence of pneumonia in any case), moderate I (no respiratory failure, pneumonia findings, or dyspnea; 93% < SpO_2_ < 96%), moderate II (respiratory failure requiring supplemental oxygen; SpO_2_ ≤ 93%), and severe (admitted to intensive care unit or requiring a ventilator).

In this study, personal information was managed using an enrollment number unique to each study subject, assigned for anonymization. A correspondence table of enrollment numbers was appropriately managed by the department in charge at the study institution providing the biological samples. In addition, materials and correspondence tables containing other personal information collected in this study were managed appropriately, in compliance with the management methods specified at the institution providing the biological samples, to protect personal information. When sharing the study’s results with institutions providing biological samples, data were handled only with the anonymized study ID assigned for this study. The only information to be provided to the collaborative research institution was the anonymized study ID, measurement data, and information without personal identification, and information that could identify a particular individual was not provided in order to prevent any possibility of risk or disadvantage associated with the leakage of information about the study subjects.

### 2.3. Sample Analyses

SARS-CoV-2 anti-spike IgG concentrations were measured at Nexelis (Laval, QC, Canada) using S-ELISA, and neutralizing activity (50% inhibitory dilution: ID_50_) was determined using the PhenoSense SARS-CoV-2 neutralizing antibody assay (PNA) at Labcorp (Indianapolis, IN, USA). The analytical methods of S-ELISA and PNA were validated by Nexelis and Labcorp, respectively.

The SARS-CoV-2 pre-spike recombinant antigen was adsorbed onto a 96-well microplate. Following incubation, the microplate was washed to remove unbound SARS-CoV-2 pre-spike recombinant antigen and blocked to prevent non-specific binding. Standards, controls, and sample dilutions were incubated in the coated microplate, in which anti-SARS-CoV-2 pre-spike IgG-specific antibodies (primary antibodies) bound to the coated antigen. Following incubation, the microplate was washed to remove unbound primary antibodies. Primary antibodies were detected with the addition of the anti-human IgG antibody (secondary antibody) conjugated to peroxidase. After incubation, the microplate was washed to remove unbound secondary antibodies. The peroxidase substrate solution, tetramethylbenzidine (TMB), was added to the microplate, and a colored product was developed that was proportional to the amount of anti-SARS-CoV-2 pre-spike IgG antibodies present in the serum sample. Then, 2N H2SO4 was added to stop the colorimetric reaction. The absorbance of each well was measured using a microplate spectrophotometer reader at a specific wavelength (450/620 nm). Antibody concentrations were calculated for each control and sample dilution by interpolation of the OD values on the 4-parameter logistic (4-PL) standard curve and adjusted according to their corresponding dilution factor. The final concentrations of controls and samples were then determined by calculating the geometric mean of all adjusted concentrations (for the given control or sample) obtained within the interpolation range of the standard curve. The mean absolute percentage of the relative error calculated from all standard points had to be 15.0% or less. Sample and control concentrations were expressed as ELISA laboratory units per milliliter (ELU/mL).

The measurement of neutralizing activity using PNA was performed by generating HIV-1 pseudovirions that expressed the SARS-CoV-2 spike protein. The reporter pseudovirus was prepared by co-transfecting HEK293 producer cells with an HIV-1 genomic vector and a SARS-CoV-2 envelope expression vector. Neutralizing antibody activity was measured by assessing the inhibition of luciferase activity in HEK293 target cells transiently expressing the ACE2 receptor following pre-incubation of the pseudovirions with the serum specimen. A serial dilution of the test serum specimen was incubated with a reporter pseudovirus to generate an inhibition curve that enabled the determination of an ID_50_ for each sample. Luminescence as an index of capability of inhibiting the virus from replicating and thereby preventing the luciferase by samples was measured as relative luminescent units (RLU). The acceptable limit of intra-assay and inter-assay precision was set <30% ID_50_ CV.

The detection limit for ELISA titers (EUL/mL) and neutralizing activity (ID_50_) was 50 and 40, respectively. Accordingly, a value of 25 (half the minimum required dilution) for ELISA and a value of 20 (half the minimum required dilution) for PNA were assigned to samples below the cutoff point.

### 2.4. Statistical Analyses

Frequency tables, number of specimens, and percentages were calculated from the categorical data. Continuous variables were expressed as the mean, median, minimum/maximum, and interquartile range. Antibody titers and neutralizing activity were stratified by age, sex, and disease severity. The geometric mean and 95% confidence interval were calculated for both S-ELISA and PNA data from eligible participants. Data analysis was performed by EPS Corporation (Tokyo, Japan), using SAS^®^ 9.4 (SAS Institute Inc., Cary, NC, USA).

### 2.5. Sample Size

In a similar study overseas [8], the correlation between the severity of COVID-19 and neutralizing antibody activities was evaluated using 32 samples from non-hospitalized individuals and 40 samples from hospitalized individuals. In this study, the sample size of at least 50 samples from non-hospitalized individuals and at least 50 samples from hospitalized individuals was set based on the number of individuals who had recuperated from previous SARS-CoV-2 infection that could be accrued by December 31, 2021 (the end date of the study period), using this overseas study report as a reference.

## 3. Results

### 3.1. Clinical Characteristics

A total of 222 patients were eligible. Of these, 72 patients were hospitalized, and 136 patients (61.3%) were male. The gender balance was similar in the non-hospitalized and hospitalized groups. The mean age of the entire cohort was 40.0 years, and the mean ages of the non-hospitalized and hospitalized groups were 36.6 and 47.1 years, respectively, suggesting that older patients were more likely to require hospitalization. Concerning the severity of SARS-CoV-2 infection, 8 patients (3.6%) were asymptomatic, 146 patients (65.8%) had mild symptoms, 39 patients (17.6%) had moderate I symptoms, and 29 patients (13.1%) had moderate II symptoms. The majority of non-hospitalized patients had mild symptoms (94.0%), whereas most hospitalized patients had moderate I or moderate II severity (52.8% and 40.3%, respectively; Table 1).

### 3.2. Immunogenicity Assessments

As S-ELISA result was deemed invalid for one patient, so the numbers of test results for S-ELISA and PNA were 221 and 222, respectively.

The geometric mean of anti-spike antibody titers and neutralizing activity were 1555 ELU/mL and 253 ID_50_ (the reciprocal serum dilution corresponding to 50% neutralization), respectively, for eligible patients (Table 2). There were no differences in either parameter according to sex. Both the anti-spike antibody titer and the neutralizing activity were approximately fourfold higher in hospitalized patients than in non-hospitalized patients, excluding patients older than 60 years. Both anti-spike antibody titers and neutralizing activity in the age group of ≥30 to <40 years was larger than that in the other two groups. No distinct differences in anti-spike antibody titer were observed between hospitalized and non-hospitalized patients in the age group of >60 years, but older hospitalized patients displayed relatively lower neutralizing activity. Both anti-spike antibody titers and neutralizing activity increased with disease severity (Table 2).

The distribution of anti-spike antibody titers by each patient is shown in Figure 1A. Seven patients (3.4%) had antibody titers lower than the detection limit (1.7; log_10_ ELISA titer), in all of whom the neutralizing activity was below the detection limit (1.6; log_10_ ID_50_), although COVID-19 had been confirmed by polymerase chain reaction or antigen test. Including these 7 patients, anti-spike antibody titers were low in 8 patients (3.6%) (log_10_ ELISA titer < 2), low–medium in 21 patients (9.5%) (2 ≤ log_10_ ELISA titer < 2.5), medium in 69 patients (31.2%) (2.5 ≤ log_10_ ELISA titer < 3), high–medium in 49 patients (22.2%) (3 ≤ log_10_ ELISA titer < 3.5), and high in 51 patients (23.1%) (log_10_ ELISA titer ≥ 3.5). Twenty-three patients developed extremely high antibody titers (log_10_ ELISA titer > 4). The gender of these 23 patients was 17 males and 6 females. Two patients aged ≥ 60 years were included in this population.

The distribution of neutralizing activity, in ascending order, is shown in Figure 1B. Twenty-six patients had neutralizing activity lower than the detection limit (1.6; log_10_ ID_50_). Of these 26 patients, 16 patients (61.5%) were between 20 and 39 years of age. When this age group was further divided into two groups, 13/16 were included in the 20–29 age group, and 3/16 were included in the 30–39 age group. Six were included in the 40–59 age group, and four were included in the over 60 years group. By severity, 2 patients were asymptomatic, 22 mild, and 2 moderate I. The antibody titers in this population were low in 8 patients, including 7 patients with titers below the detection limit of ELISA, low–medium in 12 patients (2 ≤ log_10_ ELISA titer < 2.5), and medium in 6 patients (2.5 ≤ log_10_ ELISA titer < 3).

Neutralizing activities were low in 39 patients (17.6%) (log_10_ ID _50_ < 2), low–medium in 69 patients (31.1%) (2 ≤ log_10_ ID_50_ < 2.5), medium in 46 patients (20.7%) (2.5 ≤ log_10_ ID_50_ < 3), high–medium in 28 patients (13%) (3 ≤ log_10_ ID_50_ < 3.5), and high in 9 patients (4.1%) (log_10_ ID_50_ ≥ 3.5). Four of the five patients who developed extremely high neutralizing activity (log_10_ ID_50_ > 4) were male.

The anti-spike antibody titers and neutralizing activities displayed a good correlation in both the non-hospitalized and hospitalized groups, and the correlation coefficient in all patients was 0.84 (Figure 2). Neutralizing activities gradually decreased over time after SARS-CoV-2 infection. In asymptomatic patients, serum neutralizing activity tended to disappear earlier; however, we could not find a clear relationship between the severity of SARS-CoV-2 infection and the disappearance of antibody activities. Most patients with mild or moderate symptoms had sustained neutralizing activity up to 180 days after the SARS-CoV-2 infection (Figure 3).

## 4. Discussion

In this cross-sectional study, we quantified anti-spike and neutralizing antibody levels in patients who contracted COVID-19 between January and September 2021. Accordingly, it is assumed that the dominant strains were the D614G variant from January to March, the Alpha strain from April to June, and the Delta strain from July to September [4]. In total, 96.5% (141 of 146) of patients with mild respiratory symptoms or high oxygen saturation did not require hospitalization, whereas 97% (38 of 39) of patients with moderate dyspnea or pneumonia required hospitalization, suggesting that medical care was provided appropriately during the pandemic in Japan.

It has been reported that male sex, older age, and hospitalization are associated with the anti-spike IgG response [9,10]. In the analysis stratified by age, the antibody titers in our study were lower in patients aged 60 years and older, and antibody levels did not differ between hospitalized and non-hospitalized patients in this age group, illustrating that antibodies were not produced according to disease severity and suggesting an insufficient immune response to new pathogens in older patients. Apart from this, age was identified as a negative independent variable for serum anti-SARS-CoV-2S antibody levels after immunization with mRNA-based COVID-19 vaccines, based on data from more than 2000 people [11]. Anti-spike antibody titers and neutralizing activity did not differ according to sex in this study. This result is consistent with those reported by Trinite et al. [8], who found that the plateau of neutralizing activity was similar between men and women, even though the maximum titers of neutralizing antibodies significantly differed. It is likely that there is no apparent gender difference in terms of antibody production [9,10]. However, Scully et al. reported that males are associated with a greater risk of more severe COVID-19 outcomes [12].

In this study, when patients with COVID-19 were classified by disease severity, antibody titers and neutralizing activity were low in asymptomatic and mildly ill patients, whereas they were high in patients with moderate disease severity. These results were consistent with those reports showing that severely ill patients with COVID-19 have higher anti-spike antibody levels, as well as the production of more potent neutralizing antibodies [8,13]. However, it is noteworthy that Trinité et al. reported no correlation between neutralizing capacity and length of hospitalization, indicating the possibility that the presence of neutralizing antibodies is not a determinant for the disease resolution, i.e., a contradictory situation in which neutralizing activities are not associated with clinical benefit [8]. Furthermore, it is suggested that while antibody production plays an important role in the elimination of the SARS-CoV-2 virus, the well-balanced function of both CD4+ T cells essential for antibody production and memory-B-cell formation, and CD8+ T cells providing protection against antigens, is more important for preventing aggravation [6]

Our analysis illustrated that some patients have neutralizing activity of less than 1.6 log_10_ ID_50_ despite moderate antibody levels, suggestive of the production of non-neutralizing antibodies, as well as a risk of reinfection. The neutralizing activity of each IgG fraction and the amount of SARS-CoV-2-binding antibodies in serum/plasma obtained at multiple timepoints were determined in 43 patients [14], and 16 patients with considerable antibody titers had no neutralizing activity during the observation periods. It is curious that the cohort with these characteristics mainly included patients with mild severity, and this finding was common to both studies. Further investigation regarding risk factors for the production of antibodies with insufficient activity is required in the future. Antibody levels below the detection limit were observed in younger patients in our study. The reason that this type of patient was exclusively distributed in this group is uncertain. In general, if the innate immunity adequately and strongly functioned, the virus could have been eliminated before the production of antibodies, along with functions for the aggravation of infection. Moderbacher et al. proposed the important role of higher levels of naïve T cells for eliminating viruses in the young [6]. In addition, the importance of a robust natural immune system has been pointed out, which may contribute to asymptomatic recovery despite the low production of antibodies in children and young people [7,15].

Antibody titers in patients with moderate or severe disease persist for a relatively long period (up to 180 days). Previous reports in Japan have also found that COVID-19 survivors had sustained neutralizing activity for approximately 6 or 12 months after infection [2,3]. How long the neutralizing activity of antibodies is maintained is important information for estimating the risk of reinfection. The antibody decay was analyzed by Khoury et al., and they extrapolated that the neutralizing activity would drop below the detection limit around 240 days after outbreak, while the half-life was estimated to be approximately 90 days, from which the model assumed that the decay in neutralizing activity would be the same regardless of the initial antibody titers [16]. The data concerning the antibody titers and neutralizing activity necessary for preventing infection are currently limited. However, after plasma antibodies from monkeys that had recovered from SARS-CoV-2 infection were transfused to other monkeys at various concentrations before challenge with SARS-CoV-2, the antibody titers required for protection, i.e., decreasing the amount of virus in the upper respiratory tract compared to that in the control group, was estimated at approximately 50 pseudovirus neutralizing antibody activity in the blood before infection [17]. By utilizing the data from seven clinical trials for COVID-19 vaccines and one convalescent study, Khoury et al. reported that the neutralizing activity necessary to halve the chance of infection is equivalent to about 20% of the mean neutralizing activity of convalescent plasma based on a normally distributed model, and about 30% by using a distribution-free approach [16].

The samples were cross-sectional, and the window for data collection was relatively tight. The sampled population was considered to be clinically diverse, and its wide age range is representative of the blood donor population in general clinical practice. However, this study had multiple limitations. Serum samples were not obtained from patients with severe symptoms in this study; thus, we could not fully analyze the relationship between the levels and activity of antibodies or examine the time course of antibody activity across patients with COVID-19. The data released by the Japanese government from three different districts between January and February 2022 illustrate that the rate of severe disease among infected people with no history of vaccination remains low (less than 0.5% (145/34,136 patients)) [18]. This might partially support the plausibility of sampling bias in this analysis. In addition, this study contained fewer asymptomatic patients (*n* = 8) and elderly patients (*n* = 14). Furthermore, this study consisted of samples provided by the patients at a single and arbitrary point (pooled data), rather than from sequential sampling in each patient. In interpreting the results of this stratified analysis of antibody titers and neutralizing activities, it is important to note that the time after infection, i.e., antibody decay, was not considered; therefore, this could remain as a substantial bias. The samples were collected from January to September 2021, when different variants were dominant. This might have caused a bias to the obtained antibody titers, since the antibodies’ affinity/neutralization capacity will vary with different variants. We also could not eliminate biases such as underlying factors—especially those affecting the immune system, such as antibody production—as well as alternative confounding effects associated with the baseline characteristics used in this analysis, and residual confounders.

## 5. Conclusions

Overall, this cross-sectional observational study revealed that the production of anti-spike antibodies with neutralizing activity was sustained for more than 6 months in patients who were infected with SARS-CoV-2 variants such as the Alpha and Delta variants, as well as variants carrying the D614G mutation, and also showed that there existed patients who had very low neutralizing activity (particularly in young people), as well as those who had low neutralizing activity despite a certain level of antibody titers. The results of this analysis illustrate the importance of appropriately measuring neutralizing activity and provide useful information regarding the management of COVID-19 vaccination.

## Figures and Tables

**Figure 1 viruses-15-01842-f001:**
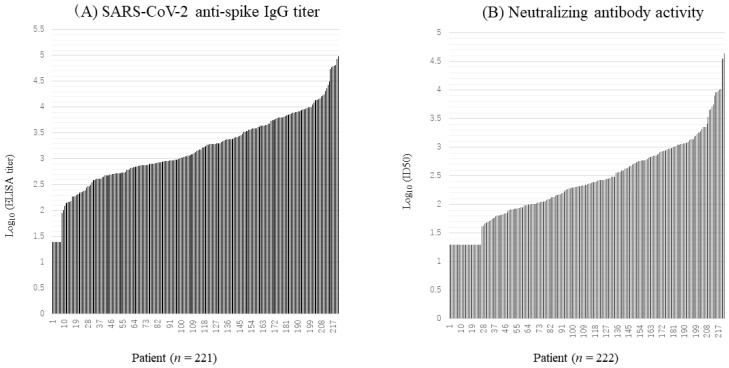
Distributions of anti-spike antibody titers (**A**) and neutralizing antibody activity (**B**) by patient. Seven patients had anti-spike antibody titers below the detection limit (1.7; log_10_ ELISA titer), and twenty-six patients had neutralizing antibody activity below the detection limit (1.6; log_10_ ID_50_). For convenience, the following criteria were applied for the levels of antibody activity and neutralizing antibody activity: low (log_10_ ID50 or ELISA titer < 2), low–medium (2 ≤ log_10_ ID_50_ or ELISA titer < 2.5), medium (2.5 ≤ log_10_ ID_50_ or ELISA titer < 3), high–medium (3 ≤ log_10_ ID_50_ or ELISA titer < 3.5), and high (log_10_ ID_50_ or ELISA titer ≥ 3.5).

**Figure 2 viruses-15-01842-f002:**
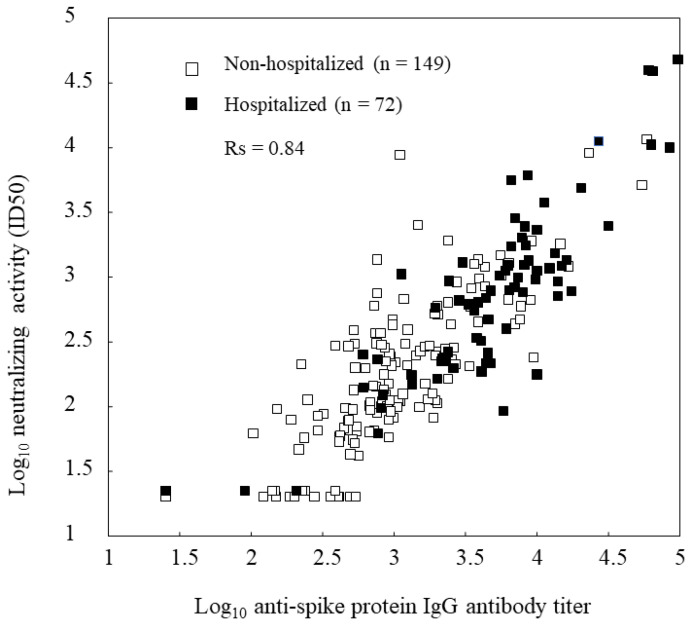
Relationship between anti-spike antibody titers and neutralizing activity; correlation coefficient was calculated excluding one patient with invalid ELISA data.

**Figure 3 viruses-15-01842-f003:**
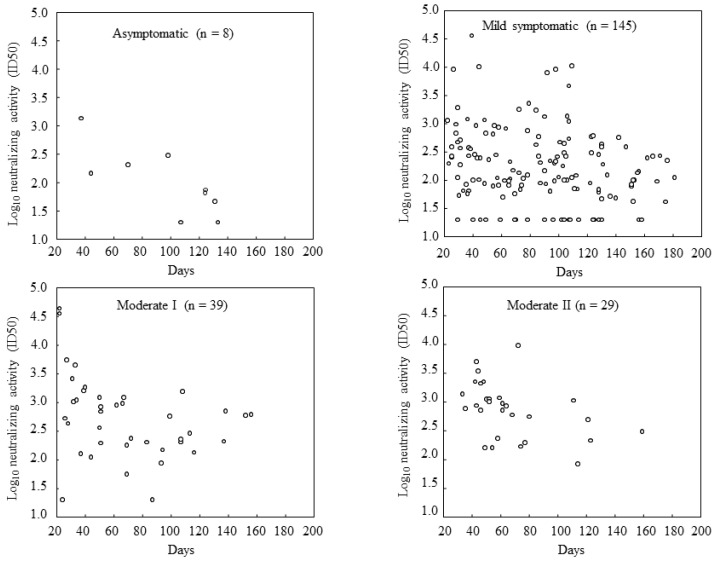
Neutralizing activity by days after infection. *X*-axis: days after the diagnosis of COVID-19. Most patients with mild or moderate symptoms had sustained neutralizing activity up to 180 days after infection. There were 26 patients with neutralizing activity below the detection limit, including 2 asymptomatic, 22 mild, and 2 moderate I cases.

**Table 1 viruses-15-01842-t001:** Patient characteristics.

	Non-Hospitalized *N* = 150 *n* (%)	Hospitalized *N* = 72 *n* (%)	Total *N* = 222 *n* (%)
Sex			
Male	92 (61.3)	44 (61.1)	136 (61.3)
Female	58 (38.7)	28 (38.9)	86 (38.7)
Age (years)			
n	150	72	222
Mean (SD)	36.6 (13.1)	47.1 (11.7)	40.0 (13.5)
Median	33.0	49.0	40.0
Min, max	20, 78	20, 79	20, 79
20 to <40 *	92 (61.3)	17 (23.6)	109 (49.1)
40 to <60	52 (34.7)	47 (65.3)	99 (44.6)
≥60	6 (4.0)	8 (11.1)	14 (6.3)
Severity of SARS-CoV-2 infection			
Asymptomatic	8 (5.3)	0	8 (3.6)
Mild	141 (94.0)	5 (6.9)	146 (65.8)
Moderate I	1 (0.7)	38 (52.8)	39 (17.6)
Moderate II	0	29 (40.3)	29 (13.1)
Severe	0	0	0

* Age of 20 to <30: total *N* = 64 (57 non-hospitalized, 7 hospitalized).

**Table 2 viruses-15-01842-t002:** Antibody titers and neutralizing activity stratified by patient characteristics between non-hospitalized and hospitalized patients.

		Antibody Titer (ELU/mL): S-ELISA			Neutralizing Activity (ID_50_): PNA		
		Non-Hospitalized	Hospitalized	Total	Non-Hospitalized	Hospitalized	Total
All		941	4391	1555	166	607	253
		(753, 1176)	(3052, 6317)	(1257, 1923)	(132, 209)	(415, 888)	(204, 313)
		***N*** = 149	***N*** = 72	***N*** = 221	***N*** = 150	***N*** = 72	***N*** = 222
Gender	Male	977	4062	1549.55	182	638	273
		(727, 1312)	(2449, 6737)	(1172, 2047)	(133, 247)	(384, 1059)	(206, 361)
		***N*** = 92	***N*** = 44	***N*** = 136	***N*** = 92	***N*** = 44	***N*** = 136
	Female	886	4963	1563	144	562	224
		(627, 1253)	(2923, 8425)	(1119, 2183)	(101, 204)	(306, 1030)	(161, 312)
		***N*** = 58	***N*** = 28	***N*** = 85	***N*** = 58	***N*** = 28	***N*** = 86
Age	20 to <40	881	2932	1065	150	375	173
(years)		(696, 1117)	(1376, 6245)	(837, 1356)	(111, 202)	(172, 817)	(131, 229)
		***N*** = 91	***N*** = 17	***N*** = 108	***N*** = 92	***N*** = 17	***N*** = 109
	40 to <60	1050	6209	2442	199	886	405
		(667, 1652)	(4173, 9240)	(1725, 3456)	(134, 295)	(561, 1399)	(291, 563)
		***N*** = 52	***N*** = 47	***N*** = 99	***N*** = 52	***N*** = 47	***N*** = 99
	≥60	983	1353	1180	159	183	172
		(80, 12,015)	(228, 8006)	(339, 4099)	(24, 1032)	(52, 636)	(70, 423)
		***N*** = 6	***N*** = 8	***N*** = 14	***N*** = 6	***N*** = 8	***N*** = 14
Severity	Asymptomatic	319	-	319	105	-	105
		(116, 877)	-	(116, 877)	(31, 353)	-	(31, 353)
		***N*** = 8	***N*** = 0	***N*** = 8	***N*** = 8	***N*** = 0	***N*** = 8
	Mild	992	4221	1043	169	610	176
		(790, 1246)	(77, 229,822)	(819, 1327)	(133, 214)	(8, 43,608)	(137, 227)
		***N*** = 140	***N*** = 5	***N*** = 145	***N*** = 141	***N*** = 5	***N*** = 146
	Moderate I	3421	3241	3245	591	521	523
		-	(1877, 5595)	(1908, 5521)	-	(302, 899)	(308, 888)
		***N*** = 1	***N*** = 38	***N*** = 39	***N*** = 1	***N*** = 38	***N*** = 39
	Moderate II		6582	6582	-	741	741
		-	(4840, 8950)	(4840, 8950)	-	(484, 1132)	(484, 1132)
		***N*** = 0	***N*** = 29	***N*** = 29	***N*** = 0	***N*** = 29	***N*** = 29

Antibody titers and neutralizing activity are expressed as the geometric mean. The 95% confidence intervals are presented within brackets. One patient was excluded from the analysis because of an invalid ELISA result.

## Data Availability

The data for this study will available upon request.

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
