# Peer review of "Acquisition of Humoral Immune Responses in Convalescent Japanese People with SARS-CoV-2 (COVID-19) Infection in 2021"

_viruses, 2023, doi:10.3390/v15091842_

Round 1
Reviewer 1 Report
1. The samples were collected from Jan to Sep 2021, when different variants were dominant. This may cause bias in the titering of antibodies and the neutralization assay. This limitation should be included in the Discussion.
2. Line 225, "2< log ELISA" should be "Log10 ELISA < 2". Line 228, "3.5<=" should be "Log10 ELISA>= 3.5". Line 229, "<4" should be "Log10 ELISA> 4"
Similar errors are found in Line 238, 248, and 251.
3. Line 223, "blow" should be "below". Line 333, delete one"are".
Author Response
We corrected critical errors in inequality signs and misspellings you kindly pointed out, and added the limitation that different variants in the observational period might cause the bias towards imminity assays.

Reviewer 2 Report
This article submitted on estimating neutralizing activity and antibody levels has high importance in managing therapeutic strategies. Although there are several reports showing humoral response post SARS-CoV-2 infection, comparision of neutralizing activity to antibody levels done in this study will be important. Also, correlation of neutralizing activity with age can predict the immune status and benefit for strategic treatment plans for patients. I strongly recommend this article for publication.
Author Response
Thank you very much for your comments.